# Influence of Aggressive Environment in Macro and Microstructural Properties of Bottom Ash Geopolymer Concrete

R. Saravanakumar [1,*], K. S. Elango [1], V. Revathi [2] and D. Balaji [3]

1 Department of Civil Engineering, KPR Institute of Engineering and Technology,
  Coimbatore 641407, Tamilnadu, India; kselango04@gmail.com
2 Department of Civil Engineering, K.S.R. College of Engineering, Tiruchengode 637215, Tamilnadu, India;
  revthiru2002@yahoo.com
3 Department of Mechanical Engineering, KPR Institute of Engineering and Technology,
  Coimbatore 641407, Tamilnadu, India; balaji.ntu@gmail.com
* Correspondence: saravanakumartg@gmail.com

**Abstract:** India generates 759.02 million metric tons of coal ash annually. Part of that quantity is successfully utilized, and the remaining portion of the ash is discarded into a landfill. There also is a need to address pollution. Cement industries are responsible for 7% of global warming. Cement has been replaced entirely by thermal power plant waste, and bottom ash is used as a binder to overcome those issues. A few researchers have carried out strength characterization, but an extensive study needs to be carried out under different environmental exposures. Therefore, the present study investigated macro and micro properties of bottom ash geopolymer concrete (BAGPC) subjected to aggressive ecological exposure conditions such as acid, salt, and sulfate attack. Sodium silicate ($Na_2SiO_3$) and sodium hydroxide (NaOH) of eight molarities were used as activators for the bottom ash geopolymer concrete (BAGPC) binder. Further bonding between steel and conventional concrete BAGPC mixes was investigated. The durability of conventional concrete (CC) was taken as the control mix to compare the durability of the optimized mix (B4) of bottom ash geopolymer. The test samples were cured for 28 days under ambient temperature and tested for the effect of $MgSO_4$, NaCl, and HCl. The strength loss and weight loss of the BAGPC B4 mix after 7, 28, 56, 90, and 180 days under aggressive conditions showed better performance than CC. It has been observed that geopolymer concrete has good bonding in nature, and the bond strength results indicate excellent bonding between steel and concrete. Microstructure studies revealed that the BAGPC B4 mix had a strong microstructure and not as much of a porous structure. It is concluded that BAGPC has potential value in the construction industry based on all aspects of the experiment.

**Keywords:** $MgSO_4$; NaCl; HCl; bond strength; microstructure; bottom ash

## 1. Introduction

India's main source of electricity is coal-based power generation. Furthermore, Indian coal is low grade, yielding 30–60% ash out of the total amount of coal. A total of 200 thermal power plants in India provided data for between April 2021 and March 2022. A total of 759.02 million metric tons of ash were generated, according to documents from the Ministry of Power, the General Electricity Authority, and the Government of India. Thermal power plants that run on coal produce a large amount of ash, which is disposed of on the nation's valuable land. The problem of air and water pollution resulting from coal-based ash has been addressed by the Ministry of Environment and Forests [1]. Due to its strength and long-lasting properties, Portland cement (PC) is a preferred binding material in concrete. Because of this, cement is widely known for rapid strength enhancement. Cement manufacture, in particular, has a detrimental influence on the environment since it uses many raw materials that consume a lot of energy and produce greenhouse gases. Calcium carbonate, limestone, and clay are nonrenewable essential ingredients used to make cement.

The cement industry contributes 7% of the planet's carbon dioxide emissions [2–4]. The sustainability of civilization is a concern for everyone, and every country is implementing the 17 Sustainable Development Goals (SDG). The year 2019 was the second warmest year on record, and carbon dioxide ($CO_2$) emissions were cited as the leading cause of global warming in SDG 13 (Climate Action). The manufacturing of Portland cement clinker results in an equivalent release of $CO_2$ into the environment [5]. Humans are consuming Portland cement alongside water in the 21st century. India stands second in the world in cement production, next to China. Thus, there is a vital need to produce novel binders that are environmentally friendly. This has sparked several investigations and studies into possibly using industrial waste products to replace cement partially. Even though industrial wastes are created in large quantities each year, only a few are used in construction projects, with the majority being dumped in landfills. The preferred use of these materials helps protect the environment from global warming, promotes sustainability, and prevents landfill waste. Still, cement replacement (partially) using industrial products will not lead to a solution to protect the globe from using cement. As a result, the world is looking for the best alternative to cement that can replace cement completely. Therefore, continuous research is being expanded worldwide.

After much research, a precious emerging cementitious geopolymer material was discovered by Prof. Davidovits. He also recognized geopolymers as a resourceful and inorganic adhesive. In 1978, Prof. Davidovits, a geopolymer pioneer, found that kaolinite aluminosilicate polymers have chemical properties that are similar to those of the materials that naturally form rocks, such as zeolites, feldspathoids, and feldspars [6]. Amorphous mineral admixtures with oxides of alumino-silicate react with alkali polysilicates to produce semi-crystalline Si–O–Al bonds. Polysialate (-Si-O-Al-O-), polysialate-silaxo (Si-O-Al-O-Si-O), and polysialate-disilaxo (-Si-O-Al-O-Si-O-Si-O-) are the three major oligomeric structures that result from the polymerization reaction. The structure of polysialate, poly sialate-siloxy, and poly sialate-disclose was proposed [7]. Fourfold coordination alumino-silicate oxides ($Si_2O_5$, $Al_2O_2$) are formed through calculations of alumino-silicate hydroxides and polymerization reactions [6–11]. The fourfold coordination of aluminum and silicate oxides with sodium or potassium hydroxide catalyst produces Si-O-Al bonds. The chemical reaction has revealed that water is used only for the workability of the geopolymer mixture, and it does not contribute anything to the chemical reaction [11]. Geopolymers use the polycondensation of silica and alumina precursors with a high alkali content to achieve strength rather than producing calcium-silicate-hydrates (CSHs) like regular Portland cement [11,12]. Many investigations have been carried out on fly ash as a source material in geopolymer concrete. Though fly ash and bottom ash are industrial by-products from thermal power plants, little work has been done on bottom ash geopolymer concrete. A few studies have only reported on the use of bottom ash either individually or with other source materials [13–16]. Due to polymerization processes, the HCl and sulfuric acid resistance of fly ash geopolymer concrete was healthier compared to concrete. Also, strength loss, weight loss, and microstructure characterization were notable in geopolymers [13] The corrosion of fly ash geopolymer concrete was evaluated in the marine environment, and sodium hydroxide sodium silicates of 8 to 14 molar concentrations with centrally reinforced beams of 12 mm diameter bars were used. Also, fly ash geopolymer specimen cracking was identified with current intensity changes. Finally, fly ash geopolymer concrete achieved excellent resistance against chloride attack compared to ordinary Portland cement concrete [17–19]. The durability of bottom ash mortar in 5% sodium sulfate and 3% sulfuric acid solutions was also investigated. The properties of the coarse medium and fine bottom ash mortar were cured at 75 °C for 48 h. The fine bottom ash had higher compressive strength and resistance against sulphate and sulfuric acid attacks [20]. The durability of a blended polymer concrete of fly ash and palm oil fuel ash immersed in 2% sulfuric acid solution for 18 months was investigated in terms of the microstructural changes, compressive strength, and mass of the combined geopolymer concrete. The aluminosilicate polymer structure present in the geopolymer concrete gave more resistance against acid

attack compared to OPC concrete [21]. The quantities of sodium hydroxide pellets and water to produce the sodium hydroxide solution per molarity were assessed. Based on Perry's handbook, sodium hydroxide solution was prepared for chemical engineers [22,23]. An empirical investigation of the function of coal bottom ash (CBA) in roller-compacted concrete was made possible by the work's concise summary of prior research on the subject [24]. A 10% mixture of CBA and glass powder showed notable increases in compressive strength, indicating that it is a viable choice for high-performance, environmentally friendly mortar [25]. Research validates the sustainability and economic viability of using calcined natural clay as a raw material for geopolymers. Tensile characteristics are improved by incorporating GGBFS and PVA fibers, providing a workable solution for resource-efficient geopolymer composites [26]. Modification of hydrophobicity reduces mesoscopic damage in geopolymers under F-T cycles in an efficient manner. By highlighting spatial gradient characteristics, CT scanning establishes the significance of the treatment in improving durability against damage caused by freezing and thawing [27].

Significant literature studies have proven that geopolymer concrete has potential value in the construction industry to minimize $CO_2$ emissions. Many researchers have also attempted to compare fly ash-based geopolymer concrete to bottom ash geopolymer concrete. Further, only some studies have reported on bottom ash geopolymer mortar and bottom ash fly ash blended concrete. Therefore, as a part of the present research, it has become a necessity to study the effect of $MgSO_4$, NaCl, HCl, bond strength, and microstructure on bottom ash geopolymer concrete.

### 1.1. Scope of the Work

Many investigations have been conducted on fly ash as source material in geopolymer concrete. Though fly ash and bottom ash are industrial by-products from thermal power plants, little work on bottom ash geopolymer concrete has been carried out. A few studies only reported on the use of bottom ash either individually or with other source materials. However, extensive studies on bottom ash in geopolymer concrete could not be explored much, especially under ambient curing conditions. Further, different environment exposures of bottom ash geopolymer concrete have yet to be investigated. Hence, an effort is been undertaken in the present work to study micro- and macrostructural properties of bottom ash geopolymer concrete.

### 1.2. Objective

○  The present work has the following important objectives;
○  To investigate performance of BAGPC under aggressive environment conditions like acid attack, salt attack, and sulphate attack;
○  To evaluate the bond strength properties of BAGPC and compare the result with control concrete (CC);
○  To identify the microstructural behavior of bottom ash geopolymer concrete (BAGPC) with ambient curing under an aggressive environment.

### 2. Materials

The present study used ordinary Portland cement (OPC) grade 53 conforming to IS 12269-1987 [28]. The physicochemical properties of the cement were tested as per IS 4031-1988 [29] and IS 12269-1987 [28]. Bottom ash (BA) was gathered from the Mettur thermal plant, Salem, Tamilnadu, India. It is an industrial by-product produced due to the burning of coal in thermal power plants [4]. The physical and chemical properties of bottom ash were studied. Bottom ash's specific gravity and surface value are 2.17 and 3460 $cm^2$/gm, respectively. The chemical composition of fly ash is shown in Figure 1. The XRD analysis was completed for the bottom ash; the test results are shown in Figure 2.

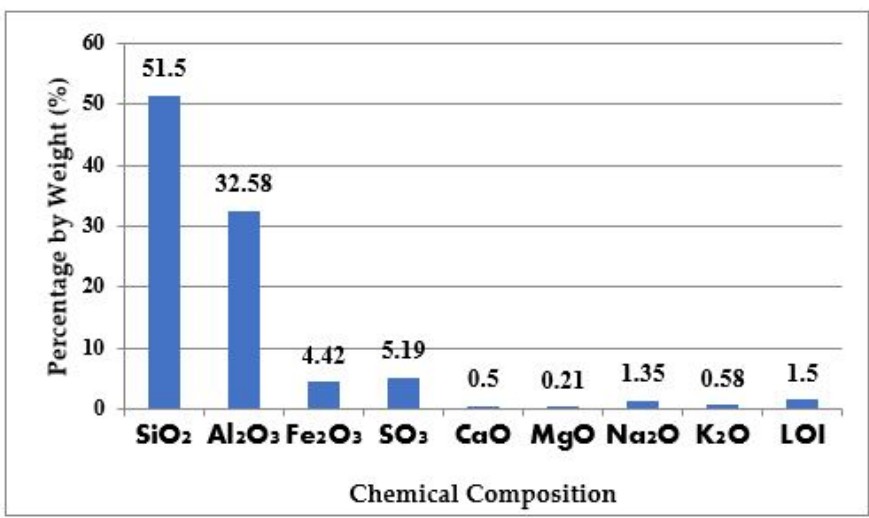

**Figure 1.** Chemical properties of the bottom ash.

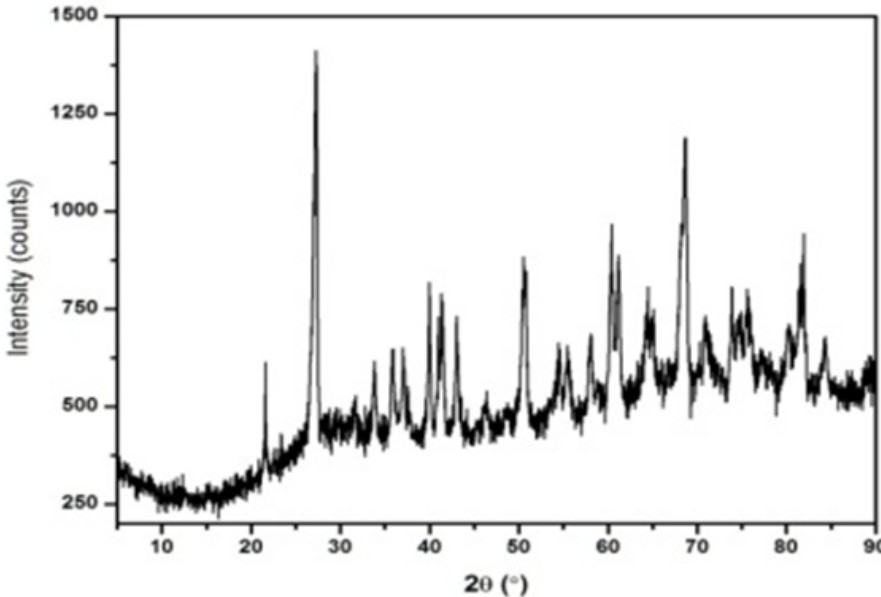

**Figure 2.** XRD of the bottom ash.

The test results show that the bottom ash had multiple peaks on the two-theta scale (ranging from 5 to 90 degrees), indicating that the BA had a polycrystalline structure. The SEM images of the bottom ash are given in Figure 3. Tiny spherical and semi-spherical particles were observed in varying sizes, and no visible pores were noticed. In this research, natural river sand was used as the fine aggregate. A sieve analysis test determined the grain size distribution as per IS 2386-1963 (Reaffirmed 2011) [30–32]. Using crushed aggregate of 20 and 12 mm in size, control and geopolymer concrete were produced as per IS 383-1970 (reaffirmed 2011) [33]. A total of 8 M sodium hydroxide in pellets were used as an alkaline activator in the geopolymer concrete. Also, sodium silicate in liquid gel was used along with sodium hydroxide solution to produce geopolymer concrete. Sulphonated naphthalene polymer Conplast SP430 superplasticizer was used in the present work, and it was purchased from FOSROC Chemicals Pvt. Ltd. (Guwahati, India). The specific gravity of the sulphonated naphthalene polymers was 1.220. Superplasticizer satisfies the recommendations of the IS code provisions.

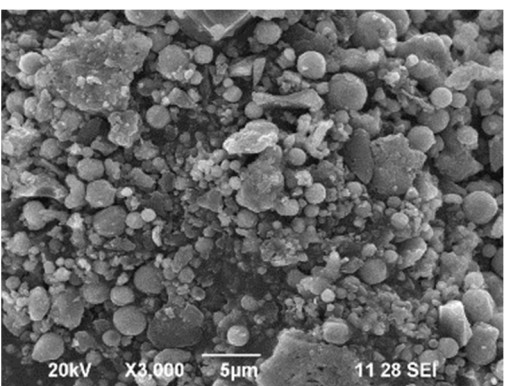

**Figure 3.** SEM image of the bottom ash.

*Mix Proportion*

Based on code IS 10262-2009, the control concrete (CC) mix design for the M40 grade was prepared [34]. Using a trial-and-error methodology, the bottom ash geopolymer concrete (BAGPC) mix was formed (Rangan 2005; Revathi et al., 2014). The molarity of the sodium hydroxide was taken as 8 M. The sodium hydroxide solution was prepared as Rajamane and Jeyalakshmi (2015) [22] recommended, and the alkaline liquid to BA ratio was kept at 0.5 on a weight basis. The proposed mixes of B4 with sodium silicate ($SiO_2$ = 29.4%, $Na_2O$ = 13.7%, water = 55.9%) to sodium hydroxide ratios of 2 on a weight basis were prepared. The final proportions of CC and BAGPC are given in Table 1. Using a pan mixer, every component was thoroughly combined to create a uniform mixture. Following two to three minutes of mixing the bottom ash and aggregates, alkaline liquid was added to the dry mix and stirred for an additional two to three minutes with the pan mixes. CC and BAGPC were mixed and then put in molds. The concrete was thoroughly crushed until the trapped air exited. After 24 h, the cast specimen was de-molded and allowed to cure for the appropriate amount of time.

**Table 1.** Proportions of various mixes.

| Mix Id | Cement kg/m³ | Bottom Ash kg/m³ | Fine Aggregate kg/m³ | Sodium Hydroxide (8 M) kg/m³ | Sodium Silicate kg/m³ | Coarse Aggregate kg/m³ | Water kg/m³ | Superplasticizer kg/m³ |
|--------|--------------|------------------|----------------------|------------------------------|-----------------------|------------------------|-------------|------------------------|
| CC | 394 | - | 629.6 | - | - | 1316.1 | 157.6 | 7.9 |
| B4 | - | 400 | 540 | 66.7 | 133.3 | 1260 | - | 8 |

## 3. Experimental Program

### 3.1. Sulphate Resistance Test

A sulfate resistance test was conducted for the control and geopolymer concretes to determine the resistance of the concrete samples against sulfate attack as per ASTM C 1012 [35]. A 100 mm × 100 mm × 100 mm sample was cast and cured for 28 days. The compressive strength of 100 mm cubes was calculated at a pace rate of $5 \pm 1$ N/mm² per second, in accordance with IS 516 [36]. At ambient temperature, the cast specimens were immersed in a 5% magnesium sulfate solution for 180 days. After 7, 28, 56, 90, and 180 days, the concrete samples were removed from the sulfate solution, and the weight and the strength loss due to the sulfate attack were measured. The weight and the strength loss of the geopolymer concrete were compared with the control concrete to assess the sulphate resistance of the geopolymer concrete. Thirty specimens were cast and subjected to a sulphate resistance test, as shown in Figure 4.

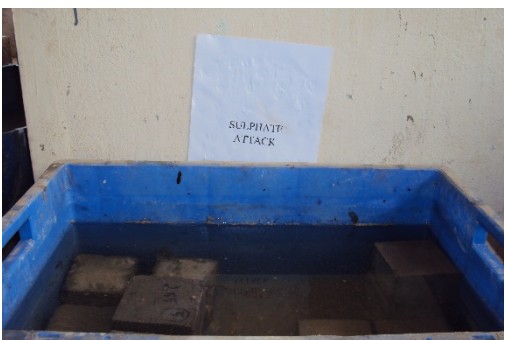

**Figure 4.** Sulphate resistance test.

### 3.2. Salt Resistance Test

Specimens 100 × 100 × 100 mm in size were cast and cured for 28 days to determine the salt resistance of the control and geopolymer concretes. The cast specimens were immersed in 3.5% NaCl solution for 180 days at ambient temperature, as shown in Figure 5. After 7, 28, 56, 90, and 180 days, the concrete samples were removed from the salt solution, and the weight of control and geopolymer concretes was determined. The percentage variation in weight and strength loss of the concrete was calculated based on the test values, and the salt resistance was determined. In total, 30 specimens were cast and tested for salt resistance.

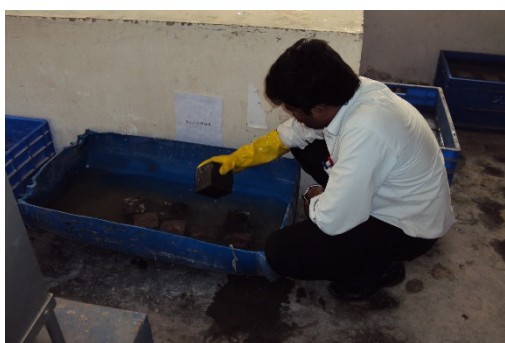

**Figure 5.** Salt resistance test.

### 3.3. Acid Resistance Test

The acid resistance of the control and geopolymer concrete specimens 100 mm × 100 mm × 100 mm in size was tested as per ASTM C 642 [37]. The cast specimens were cured for 28 days and were immersed in 3% HCl solution for 180 days at room temperature (Figure 6). After 7, 28, 56, 90, and 180 days, the concrete samples were removed from the HCl solution to determine the weight and strength loss. Acid resistance was determined based on the variations in weight and strength loss of the concrete samples. A total of 30 specimens were cast and tested for acid resistance.

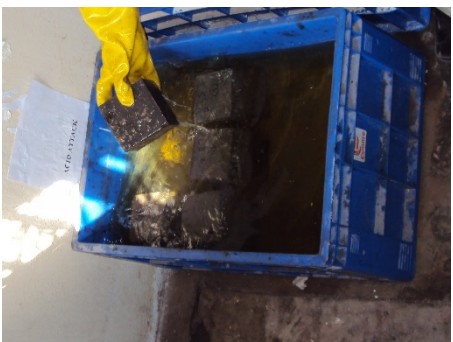

**Figure 6.** Acid resistance test.

### 3.4. Pull-out Testing

Cube-shaped concrete samples measuring 100 mm × 100 mm × 100 mm were employed for the pull-out test. The test was run by the recommended protocol outlined in IS 2770-1967 (Reaffirmed 2007) [38]. The bonding of the implanted steel with the concrete was tested on six specimens, and the load necessary for a 0.25 mm slide was noted. Using 600 kN-capacity universal testing equipment, a concrete sample was gradually loaded to evaluate the bonding stress and slip (Figure 7). Throughout the test, it was noted that the loading rate should be at most 2250 kg/min.

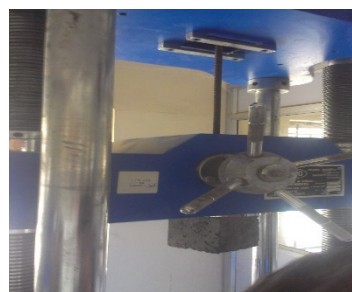

**Figure 7.** Pull-out test.

### 3.5. Scanning Electron Microscope (SEM) with Energy Dispersive X-ray Spectroscopy (EDX)

A scanning electron microscope (SEM) is the instrument most commonly used to analyze the surface morphology of a material. A scanning electron microscope consists of an electron gun, a scanning system, a column, detectors, and a substrate chamber. Electron beams are transmitted through electron guns on the surface of the materials. The transmitted beam is scattered from the material surface on the x and y axes. The detector detects the scattered electrons (secondary electrons), which are converted into signals to produce images to obtain information. Secondary electrons, X-ray photons, and secondary electrons are produced during electron beam transmission. Using energy-dispersive X-ray spectroscopy (EDX), the characteristic X-rays obtained from the refractive surface of the material are analyzed to obtain the chemical composition. The characteristic X-rays are produced due to interactions of incident electrons with the samples. Based on the measured energy, the atoms can be identified and analyzed.

## 4. Results and Discussion

### 4.1. Acid Resistance Test

The 3% hydrochloric acid solution is extensively used in chemical industries, and hydrochloric acid attack can quickly degrade industrial structures. The degree of acid attack was determined based on the strength loss and weight loss results. The weight and the strength loss of the bottom ash geopolymer concrete and the control concrete due to acid attack at 7, 28, 56, 90, and 180 days are presented in Table 2.

**Table 2.** Weight loss and strength loss of CC and BAGPC specimens under acid attack.

| Mix Id | Weight Loss (%) | | | | | Strength Loss (%) | | | | |
|---|---|---|---|---|---|---|---|---|---|---|
| | Immersion Period (Days) | | | | | Immersion Period (Days) | | | | |
| | 7 | 28 | 56 | 90 | 180 | 7 | 28 | 56 | 90 | 180 |
| CC | 1.21 | 4 | 7.3 | 9.1 | 15.4 | 1.34 | 5.23 | 10.8 | 15.15 | 21.1 |
| BAGPC B4 | 0.4 | 1.5 | 3 | 4.2 | 8 | 1 | 4.8 | 7 | 9 | 15.4 |

The comparison of the weight loss and strength loss results of the BAGPC B4 and CC specimens due to acid attack is presented in Figures 8 and 9, respectively. While analyzing the compressive strength loss of the BAGPC B4 specimens, the loss was determined to be

25%, 8.22%, 35.18%, 41.9%, and 27% lower than that of CC at the age of 7, 28, 56, 90, and 180 days, respectively. As shown in Figure 8, the weight loss of BAGPC B4 was 66.9%, 62.5%, 58.9%, 53.8%, and 48% lower than that of the control concrete at the ages of 7, 28, 56, 90, and 180 days, respectively. This shows that the bottom ash geopolymer performed better than CC because the BAGPC was porous, all aggregate pores were occupied by geopolymer paste, and the BAGPC did not have calcium compounds. Considering the distinct advantages observed in both compressive strength and weight loss for BAGPC B4 compared to CC, these findings suggest a promising potential for the utilization of bottom ash geopolymer concrete in acidic environments. The permeability and absence of calcium compounds in BAGPC contributed to its remarkable performance, opening avenues for innovative and sustainable concrete applications. Based on the test results, it is evident that BAGPC is the best alternative for CC under an aggressive environment.

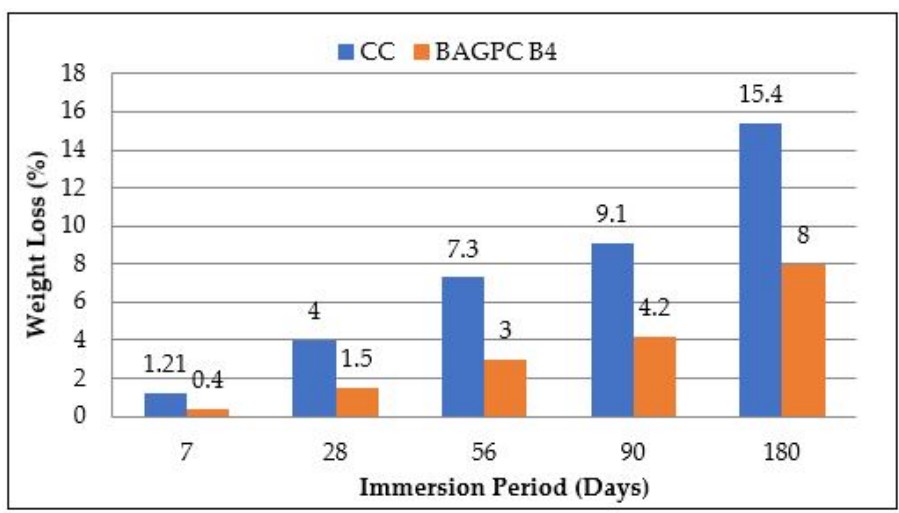

**Figure 8.** Loss of weight in CC and BAGPC specimens due to acid attack.

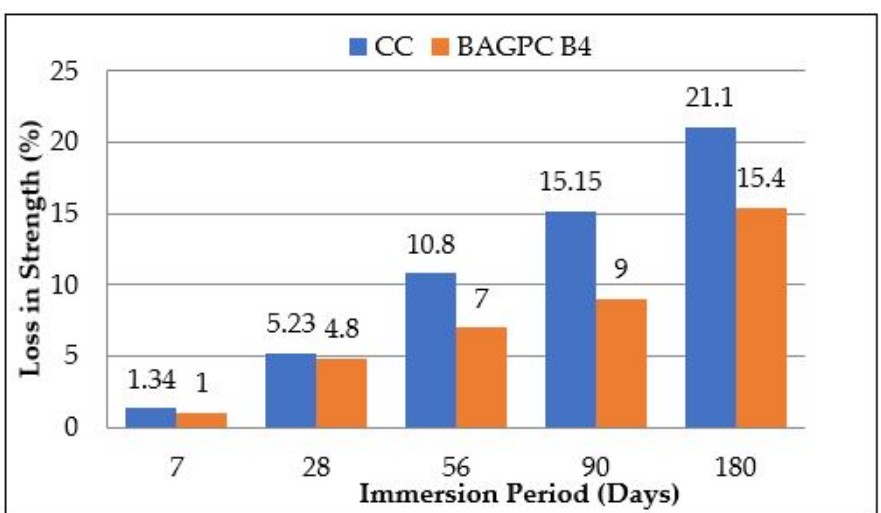

**Figure 9.** Compressive strength loss of CC and BAGPC specimens due to acid attack.

Figure 10 shows the BAGPC and CC specimens immersed in the hydrochloric acid solution for 180 days. Due to the acid attack, the CC specimen experienced surface deterioration and damage. However, the BAGPC specimen did not have any damage or deterioration. It is evident that the BAGPC specimens were less porous and had less sorptivity.

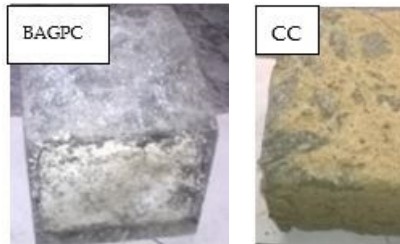

**Figure 10.** BAGPC B4 and CC specimens after 180 days immersed in acid solution.

*4.2. Sulphate Resistance Test*

The expansion and deterioration of reinforced concrete structures depend on the sulfate concentration. Generally, sulfate salts are present in aggregate, groundwater, soil, sewers, and seawater. In the present study, to analyze the strength and weight loss, BAGPC B4 and CC specimens were immersed in a 5% magnesium sulfate solution, and the test results are tabulated in Table 3.

**Table 3.** Weight loss and strength loss of CC and BAGPC specimens under sulphate attack.

| Mix ID | Weight Loss (%) | | | | | Strength Loss (%) | | | | |
|---|---|---|---|---|---|---|---|---|---|---|
| | Immersion Period (Days) | | | | | Immersion Period (Days) | | | | |
| | 7 | 28 | 56 | 90 | 180 | 7 | 28 | 56 | 90 | 180 |
| CC | 0.41 | 1.7 | 2.1 | 3 | 5.2 | 0.53 | 2.4 | 8 | 12 | 15.1 |
| BAGPC B4 | 0.1 | 0.9 | 1.7 | 2.1 | 4.7 | 0.2 | 1.7 | 3 | 6.4 | 9.8 |

The comparison of the compressive strength loss and weight loss results of BAGPC B4 and CC specimens is shown in Figures 11 and 12, respectively. As shown in Figure 12, BAGPC B4 had 62.26%, 29.16%, 62.5%, 46.6%, and 35% less compressive strength than the CC mix at 7, 28, 56, 90, and 180 days, respectively. In addition, the BAGPC mixtures exhibited excellent sulphate resistance compared to the control concrete at all ages. Further, BAGPC B4 showed 75.6%, 47%, 19%, 30%, and 9.61% less weight loss than the CC mix at 7, 28, 56, 90, and 180 days, respectively. The observed compressive strength variations between BAGPC B4 and CC suggest intricate material behavior over time. Additionally, BAGPC B4's notable resistance to sulphate-induced degradation and consistent mitigation of weight loss underscores its promise for sustainable and enduring construction applications. The visual appearances of BAGPC B4 and CC are shown in Figure 13. White precipitate was noticed on the CC specimens, but no residue was seen on the BAGPC B4 specimens because the BAGPC specimens were free of calcium compounds. The visual appearance proved that there were no dimension changes or cracks on the BAGPC or CC specimens. A similar study was carried out on bottom ash geopolymer mortar, and it was less susceptible to sulfate attack than the conventional mortar, as stated by Vanchi sata et al. [20] and Chaicharn Chotetanorm [18].

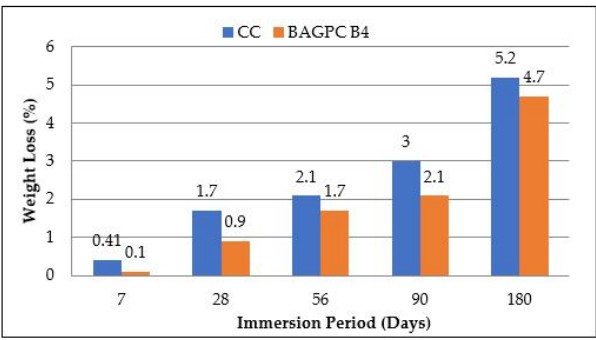

**Figure 11.** Loss of weight in CC and BAGPC specimens due to sulphate attack.

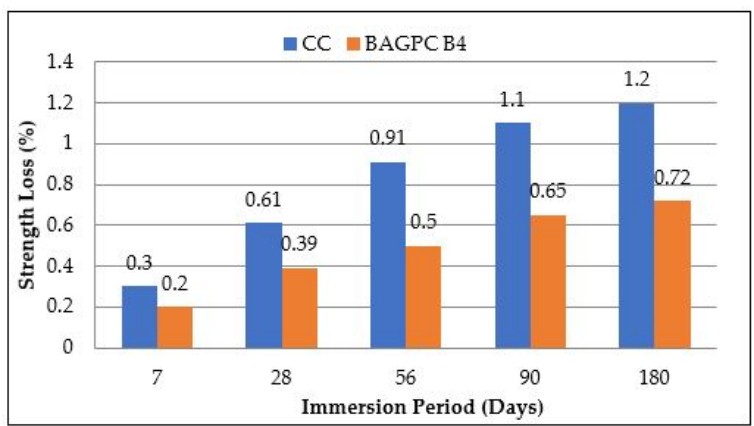

**Figure 12.** Compressive strength loss of CC and BAGPC specimens due to sulphate attack.

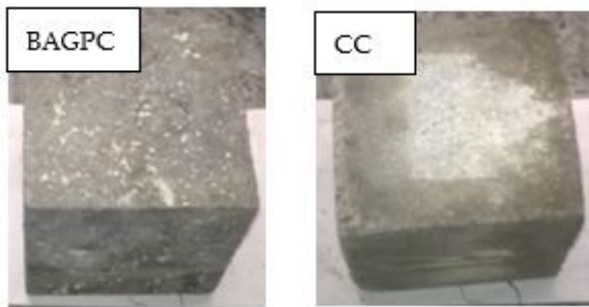

**Figure 13.** BAGPC B4 and CC specimens after 180 days immersed in sulphate solution.

*4.3. Salt Resistance Test*

Salts are present in seawater, wind from the shore, soil, and groundwater, which can affect the quality of concrete. The permissible limit of chloride for construction purposes is up to 500 ppm, as per IS 456-2000 [39]. The weight gain and the strength loss of BAGPC B4 and CC immersed in 3.5% NaCl solution at 7, 28, 56, 90, and 180 days are presented in Table 4.

**Table 4.** Weight gain and strength loss of CC and BAGPC specimens under salt attack.

| Mix Id | Weight Gain (%) | | | | | Strength Loss (%) | | | | |
|---|---|---|---|---|---|---|---|---|---|---|
| | Immersion Period (Days) | | | | | Immersion Period (Days) | | | | |
| | 7 | 28 | 56 | 90 | 180 | 7 | 28 | 56 | 90 | 180 |
| CC | 0.3 | 0.61 | 0.91 | 1.1 | 1.20 | 0.7 | 4.6 | 5.2 | 6.0 | 6.8 |
| BAGPC B4 | 0.2 | 0.39 | 0.5 | 0.65 | 0.72 | 0.1 | 2.9 | 3.7 | 4.9 | 5.4 |

The weight gain of the BAGPC B4 mix, as shown in Figure 14, was 33.3%, 36%, 45%, 40.9%, and 40% less than that of CC at 7, 28, 56, 90, and 180 days, respectively. As shown in Figure 15, BAGPC B4 showed 85.71%, 36.95%, 28.84%, 18.33%, and 20.58% less compressive stress than the mix CC at 7, 28, 56, 90, and 180 days, respectively. The weight gain of the BAGPC and CC mixes occurred due to sodium chloride. BAGPC B4 exhibited substantial reduction in weight gain compared to CC, suggesting effective mitigation of sodium chloride-induced expansion. The consistently lower compressive stress further underscores its potential in addressing expansive reactions in concrete formulations. Figure 16 shows the visual appearance of the BAGPC B4 and CC specimens immersed in a 3.5% sodium chloride solution, and it is clear that the mixes were safe against cracks, softening, and crumbling.

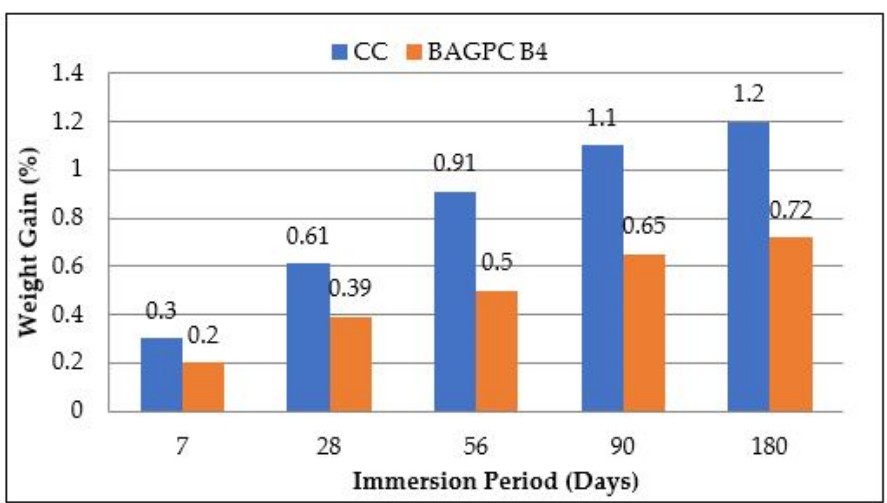

**Figure 14.** Weight gain in CC and BAGPC specimens due to salt attack.

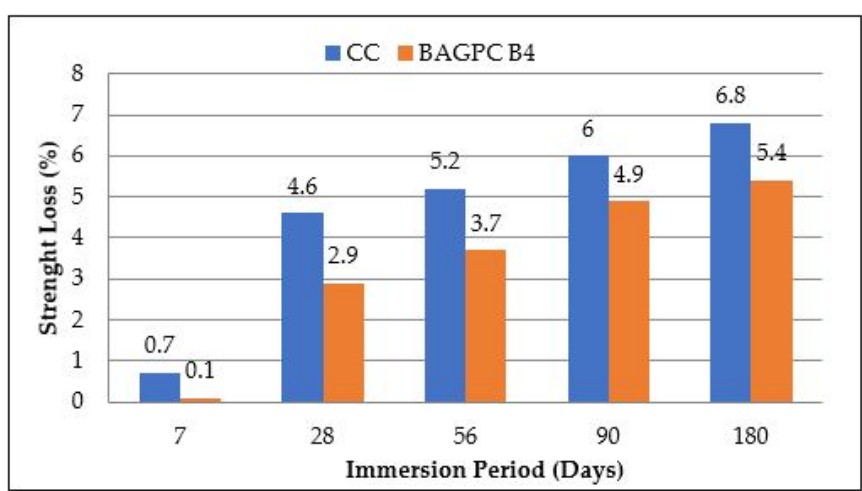

**Figure 15.** Compressive strength loss of CC and BAGPC specimens due to salt attack.

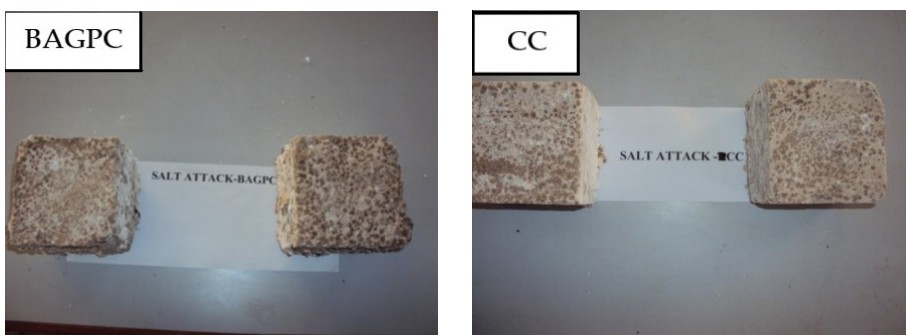

**Figure 16.** BAGPC B4 and CC specimens after 180 days immersed in salt solution.

*4.4. Pull-out Test*

The load transfer to the structure depends on the bonding between steel and cement paste. As shown in Table 5, for 28 days, the bond strength of the control mix (CC) for 0.025 mm and 0.25 mm slips was 3.29 MPa and 4.98 MPa, respectively, whereas BAGPC B4 showed 4.22 MPa and 5.61 MPa for 0.025 mm and 0.25 mm slips, respectively.

**Table 5.** Pull-out test for BAGPC and CC.

| S. No. | Description | CC | BAGPC B4 |
|---|---|---|---|
| 1 | Rebar diameter in (mm) | 12 | 12 |
| 2 | Rebar type | HYSD bars | |
| 3 | Bond strength at 0.025 mm slip (MPa) | 3.29 | 4.22 |
| 4 | Bond strength at 0.25 mm slip (MPa) | 4.98 | 5.61 |
| 5 | Design bond stress for M40 and above grade as per IS 456-2000 (MPa) | 3.04 | |
| 6 | Force required to pull out the rod (kN) | 39 | 42 |

The critical slip of 0.025 mm and the corresponding stress values were compared with IS 456-2000 design bond stress, as shown in Figure 17. The bond stress of CC and BAGPC (at 0.025 mm slip) was 8% and 38.81% higher than the design bond stress given in code IS 456-2000. Critical slip evaluations at 0.025 mm revealed that both CC and BAGPC exceeded IS 456-2000 design bond stress, emphasizing potential structural improvements in the geopolymer concrete. This result shows that both the bottom ash and the control concrete fulfilled the code requirements and that BAGPC behaved tremendously compared to CC at a critical stage. Rajamane et al. [23] (2012) reported that fly ash-based geopolymer concrete bond strength was nearly the same as control concrete, but in terms of bond stress, the bottom ash geopolymer concrete performed well compared to the control concrete. The ultimate force required to pull the rod out of BAGPC was 7.69% higher than for CC. Hence, BAGPC is recommended for all structural applications in the future construction industry.

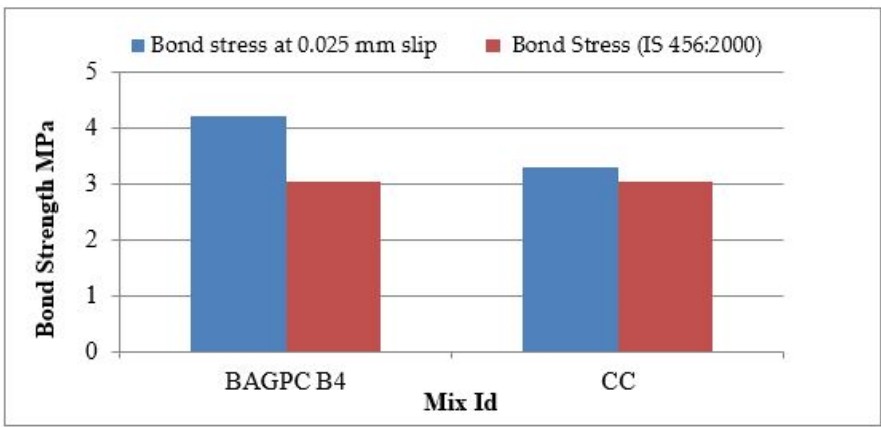

**Figure 17.** Comparison of bond strength at 0.025 mm slip vs. IS 456-2000.

*4.5. Microstructural Studies*

SEM and EDAX analysis of the B4 mix were made to evaluate the microstructure and the degree of geopolymerization element composition. The SEM and the EDAX images of the B4 mix are shown in Figures 18 and 19, respectively. The B4 mix (Figure 18b) showed a dense bottom ash geopolymer paste microstructure. Also, fewer partially and non-reacted particles, such as silica and alumina, reacted with alkaline liquid to produce thick N-A-S-H gel (Villa et al., 2010 [10]). The bottom ash geopolymer paste had no ettringite formation at 28 days in the B4 mix. The SEM of CC (Figure 18a) showed the presence of C-S-H gel and a small amount of CH. Figure 19 shows an element composition of oxygen, sodium, aluminum, silica, calcium, and iron in the B4 mix.

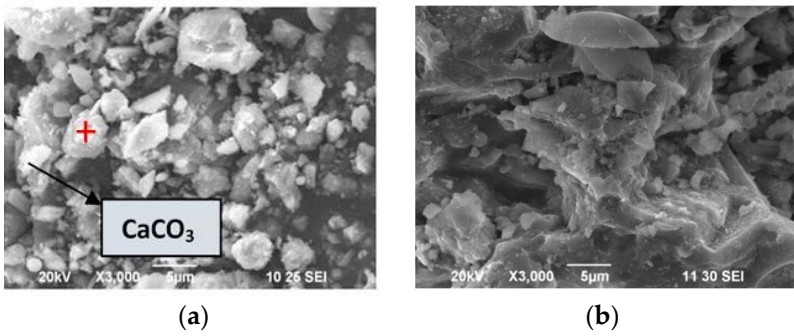

**Figure 18.** (**a**) SEM image of B4 mix. (**b**) SEM image of CC mix.

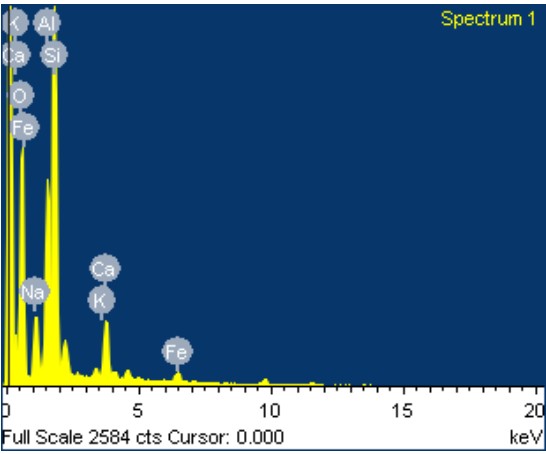

**Figure 19.** EDAX spectrum of B4 mix (+).

SEM images of the B4 mix after 180 days of immersion in 3% HCl, 5% magnesium sulphate, and 3.5% NaCl solution are presented in Figures 20–22, respectively. The B4 mix immersed in 3% HCl solution for 180 days showed fewer pores because of the leaching process of geopolymer paste aluminum and sodium in an acid solution. The B4 mix immersed in magnesium sulfate solution showed the minimum needle-shaped particles because the bottom ash geopolymer paste had a lesser Ca compound. The bottom ash geopolymer paste was less susceptible to magnesium sulphate attack than CC. The B4 mix immersed in NaCl solution exhibited strong geopolymerization of the bottom ash geopolymer paste free from pores and had no surface elimination. The element composition of O, Na, Al, Si, K, Ca, and Fe from the EDAX spectrum is presented in Figures 23–25, respectively.

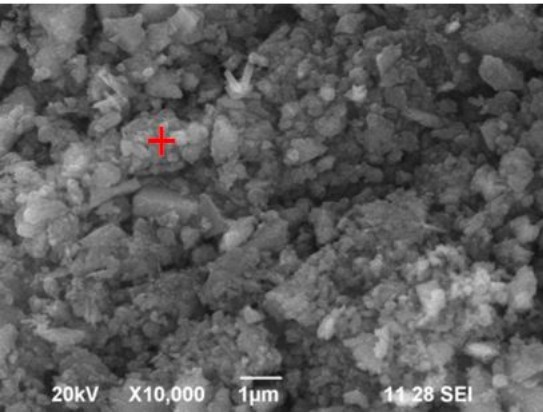

**Figure 20.** SEM image of B4 mix—3% HCl solution.

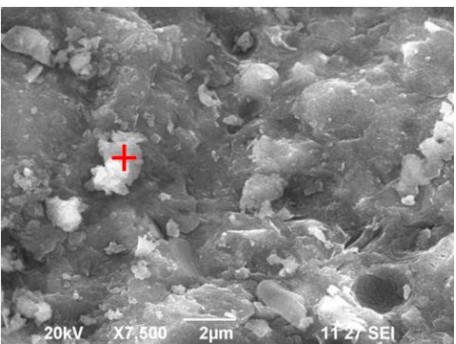

**Figure 21.** SEM image of B4 mix—5% MgSO$_4$.

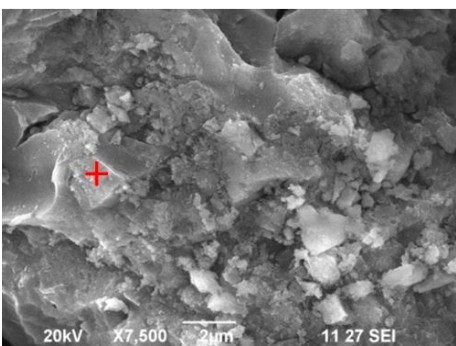

**Figure 22.** SEM image of B4 mix—3.5% NaCl solution.

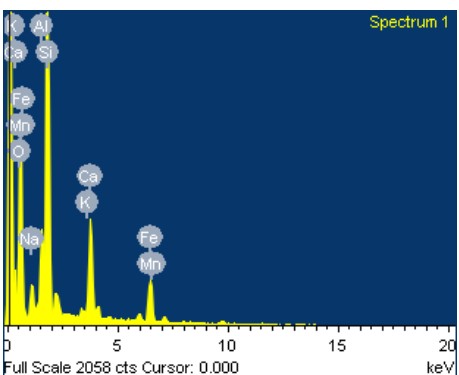

**Figure 23.** EDAX spectrum of B4 mix—3% HCl solution (+).

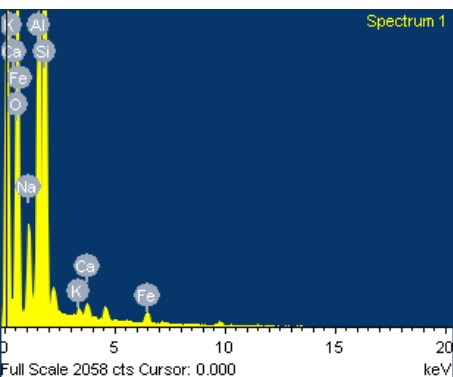

**Figure 24.** EDAX spectrum of B4 mix after 180 days immersed in 5% MgSO$_4$ solution (+).

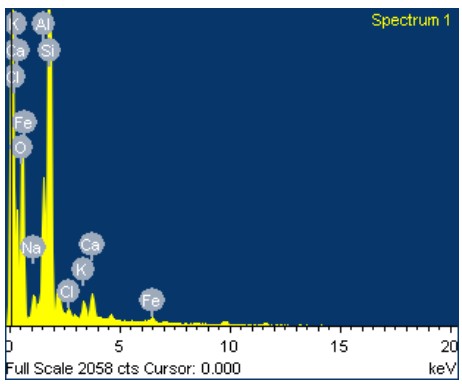

**Figure 25.** EDAX spectrum of B4 mix after 180 days immersed in 3.5% NaCl solution (+).

*4.6. Summary*

Due to acid, sulphate, and salt attacks, BAGPC B4 demonstrated lower compressive strength loss than CC at all ages because all the voids were occupied by geopolymer paste. It was also noted that the BAGPC specimens were free of cracks and crumbling, with no dimensional changes. The bond stress of BAGPC and CC were compared with the design bond stress available in IS 456-2000, and the force required to pull the reinforcement of BAGPC was 7.69% higher than for CC since geopolymer concrete has excellent bonding in nature, and these results show that BAGPC has good bonding between steel and concrete. SEM and EDAX images show that BAGPC had a dense microstructure and no ettringite formation.

**5. Conclusions**

The effect of $MgSO_4$, NaCl, HCl, bond strength, and microstructure on bottom ash geopolymer concrete were studied, and the following points were drawn.

1. The results from the acid resistance test show that the bottom ash geopolymer concrete (BAGPC) outperformed the control concrete (CC) in terms of weight loss and compressive strength. Because of its porosity and lack of calcium components, BAGPC is resistant to acid assault, making it a viable substitute for usage in acidic settings.
2. BAGPC demonstrated its exceptional sulphate resistance in the sulphate resistance test by showing much less weight loss and compressive strength loss than CC. Because BAGPC does not include calcium compounds, it is resistant to sulphates, which makes it a reliable and long-lasting choice for building applications.
3. The salt resistance test demonstrated the significant reduction in weight growth and lower compressive stress of BAGPC in comparison to CC, further highlighting its efficacy in reducing sodium chloride-induced expansion. The reliable performance of BAGPC highlights its potential to handle expansive reactions in concrete mixtures.
4. The pull-out test showed that BAGPC outperformed CC in terms of critical slip and bond strength assessments, especially the B4 mix. IS 456-2000 design bond stress was exceeded by both CC and BAGPC, with BAGPC performing better. This supports the proposal of BAGPC for a variety of structural applications in the building sector and recommends it for structural enhancements.
5. Microstructural examinations using SEM and EDAX revealed that BAGPC's geopolymer paste microstructure was dense and well formed. The durability and resistance of BAGPC to harsh environments were further reinforced by the lack of ettringite production, the presence of robust geopolymerization in NaCl solution, and the limited number of needle-shaped particles in magnesium sulphate solution.
6. All things considered, the thorough testing and analysis confirmed that bottom ash geopolymer concrete—in particular, the B4 mix—demonstrated exceptional durability, resistance to harsh chemicals, and improved structural qualities, rendering it a sustainable and advantageous option for a range of construction applications.

**Author Contributions:** Conceptualization, K.S.E.; Writing—original draft, R.S.; Writing—review & editing, D.B.; Supervision, V.R. All authors have read and agreed to the published version of the manuscript.

**Funding:** This research received no external funding.

**Informed Consent Statement:** Not applicable.

**Data Availability Statement:** The data presented in this study are available on request from the corresponding author.

**Conflicts of Interest:** The authors declare no conflicts of interest.

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
