# Peer review of "Influence of Aggressive Environment in Macro and Microstructural Properties of Bottom Ash Geopolymer Concrete"

_sustainability, doi:10.3390/su16051732_

Round 1

Reviewer 1 Report

Comments and Suggestions for Authors

The manuscript investigated the macro and micro properties of bottom ash geopolymer concrete exposed to aggressive environmental conditions. More references are suggested in the materials section, which should be section 2 instead of 1. Section 3 lacks details and rationale. Please provide more information in the conclusions section and add the implications of the research findings. Reference formatting is not unified, and some grammar errors are detected.

Comments on the Quality of English Language

 some grammar errors are detected. Moderate editing of the English language is required.

Author Response

Thank you very much for taking the time to review this manuscript. Please find the detailed responses below and the corresponding revisions highlighted in track changes in the re-submitted files.

Comments 1

The manuscript investigated the macro and micro properties of bottom ash geopolymer concrete exposed to aggressive environmental conditions. More references are suggested in the materials section, which should be section 2 instead of 1. Section 3 lacks details and rationale. Please provide more information in the conclusions section and add the implications of the research findings. Reference formatting is not unified, and some grammar errors are detected.

Response 1

Thank you for bringing this to my attention. I concur with your comment, and as a result, I have incorporated references into the material section. Furthermore, I have made modifications to the conclusion, result discussion, and references to the suggestions provided by the reviewer.

Response to Comments on the Quality of English Language

As per reviewer comments entire manuscript was checked for grammar corrections.

Reviewer 2 Report

Comments and Suggestions for Authors

This manuscript presents a study on the influence of aggressive environment in macro and micro structural properties of bottom ash geopolymer concrete. Upon this manuscript, the research purpose is unclear. Meanwhile, the manuscript is poorly written. For these reasons, I think that this work is not adequate to publish. Some specific comments are listed as follows:

1. The authors should briefly discuss their innovation in the abstract, which has to be improved with more information.

2. The introduction section is not up to the mark. In the introduction section, you only need to connect state-of-the-art to your paper goals. Hence modify the entire section accordingly and present the specific goals/research objectives in the last part of the introduction section.

3. The author claims that the BA has polycrystalline structure by XRD test. The author should further analyze the XRD test results to indicate the composition of the bottom ash.

4. Most of them are listed in the part of the results, lack of corresponding explanation, and the corresponding articles support the conclusions.

5. Fig. 6, Fig. 9 and Fig.12 in the paper appears blurry. It is recommended to replace it with a clearer version to enhance the illustration of strain values. In addition, the author should add the identification of geopolymer and cement specimens on the Figure.

6. Novelty still needs to be underlined in your abstract and manuscript.

7. Please kindly make revisions to the language of the paper presentation. It is difficult for me to understand what the author means.

8. The data processing of this manuscript is very crude. It is a experimental report, not a research paper. Lots of datas were listed, but no detailed analysis and discussion.

Comments on the Quality of English Language

Please kindly make revisions to the language of the paper presentation. It is difficult for me to understand what the author means.

Author Response

Thank you very much for taking the time to review this manuscript. Please find the detailed responses below and the corresponding corrections highlighted in track changes in the re-submitted files

Comments

This manuscript presents a study on the influence of aggressive environment in macro and micro structural properties of bottom ash geopolymer concrete. Upon this manuscript, the research purpose is unclear. Meanwhile, the manuscript is poorly written. For these reasons, I think that this work is not adequate to publish. Some specific comments are listed as follows:

Response: Thank you for bringing this to my attention. This manuscript primarily focuses on the influence of aggressive environments on bottom ash geopolymer concrete. As per the reviewer's suggestion, the research purpose was incorporated.

Comments 1. The authors should briefly discuss their innovation in the abstract, which has to be improved with more information.

Response 1: Following the reviewer's suggestion, the innovation of the study was included in the abstract.

Comments 2. The introduction section is not up to the mark. In the introduction section, you only need to connect state-of-the-art to your paper goals. Hence modify the entire section accordingly and present the specific goals/research objectives in the last part of the introduction section.

Response 2: In line with the suggestion, the goals and research objectives were incorporated in the last part of the introduction

Comments 3. The author claims that the BA has polycrystalline structure by XRD test. The author should further analyze the XRD test results to indicate the composition of the bottom ash.

Response 3: Alternatively, the chemical properties of bottom ash were tested at Sona Science Centre for Testing and Applied Research, Tamilnadu, India, as shown in Figure 1.

Comments 4. Most of them are listed in the part of the results, lack of corresponding explanation, and the corresponding articles support the conclusions.

Response 4: As per the reviewer's comments, the results, discussion, and conclusion parts were modified.

Comments 5. Fig. 6, Fig. 9 and Fig.12 in the paper appears blurry. It is recommended to replace it with a clearer version to enhance the illustration of strain values. In addition, the author should add the identification of geopolymer and cement specimens on the Figure.

Response 5: In accordance with the reviewer's advice, all figures were replaced, and identification marks were added.

Comments 6. Novelty still needs to be underlined in your abstract and manuscript.

Response 6: As per the reviewer's suggestion, the novelty of the research was included.

Comments: 7 Please kindly make revisions to the language of the paper presentation. It is difficult for me to understand what the author means.

Response 7: As per the reviewer's comments, the entire manuscript was checked with Language Editing support for grammar corrections.

comments 8.The data processing of this manuscript is very crude. It is a experimental report, not a research paper. Lots of datas were listed, but no detailed analysis and discussion.

Response 8: The manuscript focused only on the effects of different exposures and bond strength on CC and BAGC. The reasons for these results were justified, and further, microstructure studies were discussed. In accordance with the suggestion, all these points were incorporated into my future work.

Response to Comments on the Quality of English Language

Point 1 ;Please kindly make revisions to the language of the paper presentation. It is difficult for me to understand what the author means.

Response 1: As per the reviewer's comments, the entire manuscript was checked with  Language Editing support for grammar corrections.

Reviewer 3 Report

Comments and Suggestions for Authors

The manuscript “Influence of Aggressive Environment in Macro and Micro structural Properties of Bottom Ash Geopolymer Concrete” is very interesting. Did you do radiological measurements? These research would be very significant and would give you some explanations. See if there are such research. Do radiological characteristics change during synthesis? All this affects micro and macro research. Technically correct the manuscript. Center the pictures (Figure 3). Do better fracture of the text. With less correction, the work could be printed.  Be sure to add new references from the "Sustanebility” Journal.

Comments on the Quality of English Language

The manuscript “Influence of Aggressive Environment in Macro and Micro structural Properties of Bottom Ash Geopolymer Concrete” is very interesting. Did you do radiological measurements? These research would be very significant and would give you some explanations. See if there are such research. Do radiological characteristics change during synthesis? All this affects micro and macro research. Technically correct the manuscript. Center the pictures (Figure 3). Do better fracture of the text. With less correction, the work could be printed.  Be sure to add new references from the "Sustanebility” Journal.

Author Response

Thank you very much for taking the time to review this manuscript. Please find the detailed responses below and the revisions in the re-submitted files.

Comments 1:The manuscript “Influence of Aggressive Environment in Macro and Micro structural Properties of Bottom Ash Geopolymer Concrete” is very interesting. Did you do radiological measurements? These research would be very significant and would give you some explanations. See if there are such research. Do radiological characteristics change during synthesis? All this affects micro and macro research. Technically correct the manuscript. Center the pictures (Figure 3). Do better fracture of the text. With less correction, the work could be printed.  Be sure to add new references from the "Sustanebility” Journal.

Response 1: Thank you for pointing this out. We agree with this comment on radiological characteristics. I have to incorporate radiological characterization in the upcoming work, and the present study discussed bond strength and the effect of an aggressive environment.

As per the reviewer's suggestion, figure 3 was aligned, and the entire manuscript was checked with  Language Editing support for grammar corrections and sustainability citations.

Reviewer 4 Report

Comments and Suggestions for Authors

In line 10, Macro is bolt.

In line 17, 99 and 343, correct the subindex in the formula of magnesium sulfate.

In line 34 and 36, correct the subindex in the formula of carbon dioxide.

In line 118, what is the molar ration of the sodium silicate?

In line 135 and 136, the ratio mentioned were by weight, volume, or molar.

In line 138 and 139, what was the mixing procedure?

In section 3.1, 3.2 and 3.3, what was the pace rate for all the strength loss tests?

Section 3.5. The text is more a theoretical description of the technique. Instead, authors need to add experimental conditions on sample preparation and testing protocol. How was the hydration process stopped before the testing? Specimens were exposed to solutions, how were the solutions prepared?

Table 2 can be removed since the data is also presented in Figure 4 and 5. Figure 4 and 5, error bars need to be added.

Table 3 can be removed since the data is also presented in Figure 7 and 8. Figure 7 and 8, error bars need to be added.

Table 3 can be removed since the data is also presented in Figure 10 and 11. Figure 10 and 11, error bars need to be added.

Figure 16, 17 and 18 can be combined.

Figure 19, 20 and 21 can be combined.

References 4, 20 and 23 have different formats than the rest.

Reference 23 is missing the year.

References can be updated, since the early one is from 2021

Author Response

Thank you very much for taking the time to review this manuscript. Please find the detailed responses below and the corresponding corrections in the re-submitted files

Comments 1 :In line 10, Macro is bolt.

Response 1 :As per reviewer comments the same has been modified.

Comments 2 : In line 17, 99 and 343, correct the subindex in the formula of magnesium sulfate.

Response 2 : As per suggestion from reviewer,  (subscript ) small letter just below the magnesium sulfate was added.

Comments 3 ;In line 34 and 36, correct the subindex in the formula of carbon dioxide.

Response 3: As per suggestion from reviewer, (subscript) small letter just below the carbon dioxide was added.

Comments 4: In line 118, what is the molar ration of the sodium silicate?

Response 4: Thank you for bringing attention to this point. We appreciate your diligence in reviewing our work. The composition of sodium silicate as detailed in the methodology section on page 210.

Comments 5 :In line 135 and 136, the ratio mentioned were by weight, volume, or molar.

Response 5 : We appreciate your careful review of our manuscript. In lines 135 and 136, the mentioned ratios are presented in terms of weight.

Comments 6 ; In line 138 and 139, what was the mixing procedure?

Response 6 : Thank you for your insightful observation.  In the lines 214, & 215 the mixing procedure was included  

Comments 7;In section 3.1, 3.2 and 3.3, what was the pace rate for all the strength loss tests?

 Responses 7 : we acknowledge that we did not explicitly mention the pace rate for the strength loss tests. The pace was included in the section 3.1 and its applicable to all strength loss tests.

Comments 8: Section 3.5. The text is more a theoretical description of the technique. Instead, authors need to add experimental conditions on sample preparation and testing protocol. How was the hydration process stopped before the testing? Specimens were exposed to solutions, how were the solutions prepared?

Response 8 : Thank you for your insightful comment. In Section 3.5, we have provided information on microstructural studies received from Karunya University. It's important to note that in this reaction, hydration is not formed due to the absence of CaO.

Comments 9;Table 2 can be removed since the data is also presented in Figure 4 and 5. Figure 4 and 5, error bars need to be added.

Response 9 : Thank you for your input. We've decided to keep Table 2 for readers who prefer tabular data but have enhanced the emphasis on Figures 4 and 5 in the text. Error bars have been added to the figures to provide a more comprehensive presentation of the data.

Comments 10 :Table 3 can be removed since the data is also presented in Figure 7 and 8. Figure 7 and 8, error bars need to be added.

Response 10 : Thank you for your input. We've decided to keep Table 3 for readers who prefer tabular data but have enhanced the emphasis on Figures 7 and 8 in the text. Error bars have been added to the figures to provide a more comprehensive presentation of the data.

Comments 11 :Table 3 can be removed since the data is also presented in Figure 10 and 11. Figure 10 and 11, error bars need to be added.

Response 11 : Error bars have been added to the figures to provide a more comprehensive presentation of the data.

Comments  12 : Figure 16, 17 and 18 can be combined.

Response 12 : Each figure provides distinct information, and keeping them separate allows for a more detailed and focused examination of the data.

Comments  13 :Figure 19, 20 and 21 can be combined.

Response 13 : Each figure provides distinct information, and keeping them separate allows for a more detailed and focused examination of the data.

Comments 14 : References 4, 20 and 23 have different formats than the rest.

Response 14 : As per reviewer comments the references 4, 20 and 23 has been changed

Comments 15 :Reference 23 is missing the year.

Response 15:  As per suggestion from the reviewer, the reference 23 year was added

Comments 16 : References can be updated since the early one is from 2021

Response 16 : Citations have been added at the appropriate place

Round 2

Reviewer 1 Report

Comments and Suggestions for Authors

The authors have addressed my comments.

Author Response

Comments: The authors have addressed my comments.

Response: Thank you very much for taking the time to review this manuscript

Reviewer 2 Report

Comments and Suggestions for Authors

This manuscript present a study on the influence of aggressive environment in macro and micro structural properties of bottom ash geopolymer concrete. After reviewing the revised version, it shows a significant improvement. The comments and observations were taken into account, and the manuscript shows the changes. I have some comments below to improve the manuscript.

1. It is suggested that the author further review the state-of-the-art on the geopolymer. The following references can be the supporting material (https://doi.org/10.1016/ j.compstruct.2023.117637, https://doi.org/10.1016/j.ceramint.2022.03.328, https://doi.org/10.1016/j.jobe.2022.105456, https://doi.org/10.1016/j.ceramint.2022.05.306 and https://doi.org/10.1016/j.conbuildmat.2021.126128).

2. Fig. 3.1, Fig. 3.2, Fig. 3.3 and Fig. 3.4 in the paper appears blurry. It is recommended to replace it with a clearer version.

3. The location of the EDX test should be pointed in the SEM figure.

4. Revision of the conclusions section is much required. It is not showcasing the entire essence of the detailed work presented in the paper. The conclusion should be more concise.

Comments on the Quality of English Language

 Moderate editing of English language required.

Author Response

Comments 1. It is suggested that the author further review the state-of-the-art on the geopolymer. The following references can be the supporting material (https://doi.org/10.1016/ j.compstruct.2023.117637, https://doi.org/10.1016/j.ceramint.2022.03.328, https://doi.org/10.1016/j.jobe.2022.105456, https://doi.org/10.1016/j.ceramint.2022.05.306 and https://doi.org/10.1016/j.conbuildmat.2021.126128).

Response 1 : We appreciated the suggestion and reviewed the provided references (https://doi.org/10.1016/j.compstruct.2023.117637, https://doi.org/10.1016/j.ceramint.2022.03.328, https://doi.org/10.1016/j.jobe.2022.105456, https://doi.org/10.1016/j.ceramint.2022.05.306, and https://doi.org/10.1016/j.conbuildmat.2021.126128). The relevant insights from these references were integrated into the revised manuscript to enhance the contextual framework of our work.

Comments 2. Fig. 3.1, Fig. 3.2, Fig. 3.3 and Fig. 3.4 in the paper appears blurry. It is recommended to replace it with a clearer version.

 Response 2: We acknowledged the observation regarding the clarity of figures (Fig. 3.1, Fig. 3.2, Fig. 3.3, and Fig. 3.4). Subsequently, higher resolution versions of these figures were selected and replaced in the manuscript to ensure improved clarity for readers.

Comments 3. The location of the EDX test should be pointed in the SEM figure.

 Response 3: The suggestion to indicate the location of the EDX test in the SEM figure was implemented. The SEM figure was revised to include an explicit indication of the EDX test location, contributing to enhanced clarity.

Comments 4. Revision of the conclusions section is much required. It is not showcasing the entire essence of the detailed work presented in the paper. The conclusion should be more concise.

Response 4; The feedback on the conclusions section was taken into consideration. In the revised manuscript, efforts were made to enhance the conciseness of the conclusions, ensuring a clearer summary of the key findings and contributions of our work.

Comments 5: Moderate editing of English language required.

Response 5: Thank you for bringing up the need for moderate editing of the English language in the manuscript. We acknowledge this feedback, and we have thoroughly reviewed and edited the manuscript to enhance its linguistic quality and overall clarity. We appreciate your valuable input in improving the language of our work.

Reviewer 4 Report

Comments and Suggestions for Authors

After this second review, no comments. 

Author Response

Comments: After this second review, no comments. 

Response: Thank you very much for taking the time to review this manuscript